# Impact of Different Isokinetic Movement Patterns on Shoulder Rehabilitation Outcome

**DOI:** 10.3390/ijerph191710623

**Published:** 2022-08-25

**Authors:** Martin Missmann, Katrin Gollner, Andrea Schroll, Michael Pirchl, Vincent Grote, Michael J. Fischer

**Affiliations:** 1Austrian Workers’ Compensation Board AUVA, 1201 Vienna, Austria; 2VAMED Rehabilitation Center Kitzbuehel, Hornweg 32, 6370 Kitzbuehel, Austria; 3Department of Internal Medicine II, Medical University of Innsbruck, 6020 Innsbruck, Austria; 4Ludwig Boltzmann Institute for Rehabilitation Research, Kurbadstrasse 14, 1100 Vienna, Austria

**Keywords:** shoulder, rehabilitation, isokinetic training, DASH questionnaire

## Abstract

Shoulder pain is regularly associated with limited mobility and limitations in activities of daily living. In occupational therapy, various interventions, including active isokinetic training with a Baltimore Therapeutic Equipment (BTE) Work Simulator, help the patient improve shoulder mobility and alleviate pain. This randomized controlled cohort study aims to evaluate the impact of different isokinetic movement patterns on the DASH score, pain, and objective performance measures, such as range of motion (ROM) and hand grip strength. Patients that participated in a specific 3-week inpatient orthopedic rehabilitation were divided into two groups. The first group (UNI-group, n = 9) carried out uniplanar exercises for shoulder flexion, abduction, and external rotation. The patients in the second group (ADL-group, n = 10) imitated multiplanar everyday movements, such as climbing on a ladder, loading a shopping cart, and raising a glass to their mouth. Compared to the UNI-group, the ADL-group improved significantly in DASH scores (mean −10.92 ± 12.59 vs. −22.83 ± 11.31), pain (NPRS −1.11 ± 2.37 vs. 3.70 ± 2.00), and shoulder abduction (+2.77 ± 15.22 vs. +25.50 ± 21.66 degrees). In conclusion, the specific BTE exercise program with multiplanar movement patterns contributed considerably to the therapeutic improvement.

## 1. Introduction

Shoulder pain is one of the most common complaints of the musculoskeletal system, with a lifetime prevalence estimated between 7 and 10%, and is most frequently caused by impingement syndromes and rotator cuff tears [1,2]. Other causes for shoulder pain should also be considered in the differential diagnosis [3], such as osteoarthritis, nerve irritation [4], instability, Parsonage–Turner syndrome [5,6], tumors, and conditions of inner organs [7,8,9,10], rheumatic diseases [11], alterations of local blood vessels [12], trauma, and osteonecrosis [13].

Precise knowledge of anatomy and the functional status of a shoulder disorder is crucial for correct diagnosis and therapy planning. Shoulder pain is commonly associated with discomfort and limitation of shoulder function, range of motion (ROM), and muscular strength. Chronic pain has a major impact on physical, emotional, and cognitive functioning [14,15] and is assessed by patients themselves by means of rating scales such as Numerical Pain Rating Scales (NPRS) in the present study. The NPRS is used to describe the intensity of pain by means of a 0–10 scale, with zero meaning “no pain” and ten meaning “the worst pain imaginable” [16].

Objective measurements captured by clinicians and therapists allow conclusions to be drawn about the extent to which the movements are carried out and with what force. The measurement of grip strength has been widely adopted as a singular indicator of overall strength [17,18]. Reduced grip strength is associated with impaired quality of life in older adults and is an established marker of frailty, predicting physical decline and functional limitation in daily living [19]. In addition, conditions of the elbow, forearm, or wrist influence the grip strength ratio and the results of the DASH [20,21]. In the present study, a Jamar hydraulic dynamometer was used for measuring grip strength [22]. A number of tools have been designed to measure joint range of motion, varying from simple visual estimation to high-speed cinematography, using a conventional goniometer, digital devices, or a radiographic joint angle measurement [23,24,25]. As in the present study, the universal fullcircle goniometer is the preferred instrument for measuring the ROM [26,27,28].

The shoulder complex is an arrangement of anatomical structures that facilitate the functional ROM of the upper extremity [29]. Therapy should be tailored to the specific causes; for example, internal specialists should treat internal diseases that radiate into the shoulder. Based on the recommendations for the therapy of shoulder pain, the type of treatment should be centered on physical assessment findings.

The goal of treatment is to reduce pain and improve ROM to restore shoulder function [30]. Depending on the type and state of shoulder conditions, surgery may be necessary and steroids or nonsteroidal anti-inflammatory drugs (NSAIDs) may be prescribed for their anti-inflammatory effects. Physical and manual therapy can be applied, and instructions for active and passive movement therapy can be given. Active exercises should be the primary treatment approach, containing a limited number of exercises, without provoking the presenting shoulder pain. Klintberg and colleagues recommend principles for an exercise program, such as increasing difficulty and intensity and progression from basic (simple) to more functional (complex) shoulder movements [31].

Therapeutic exercise patterns frequently relate to daily life movement patterns [3,32]. In today’s occupational therapy interventions, innovative and computer-aided devices have increasingly replaced traditional methods [33]. In the present study, we describe the comparison of two exercise options on the Baltimore Therapeutic Equipment (BTE) Work Simulator, using different movement patterns for each group. First, standard movements in one plane for the group are referred to as the uniplanar or “UNI-group”. Second, complex multiplanar movement patterns are referred to as the “ADL-group”, imitating activities of daily living. This retrospective randomized single-center pilot study examines two different interventions and their effects on the results of the DASH questionnaire, pain, hand grip strength, and shoulder mobility. The aim of the study was to evaluate the differences between the two groups in patient-reported outcome measures (PROMs) and objective measures.

## 2. Materials and Methods:

### 2.1. Study Design

A consecutive enrollment of 117 inpatients with shoulder complaints in a specialized orthopedic rehabilitation facility was assessed for eligibility for this randomized controlled cohort study. Out of the initially recruited patients, 97 did not meet the inclusion criteria. The exclusion criteria were restricted shoulder movement due to a bandage or cast, previously undergone shoulder surgery, psychiatric illnesses or cognitive impairment, open wounds, or acute inflammatory processes. In addition, patients with habitual multidirectional instabilities or a neurologically effective disc herniation of the cervical and upper thoracic spine were excluded. After detailed clarification, 20 patients remained for the initial examination took place. The group allocation was randomized for 20 patients, 10 in each group. There was a dropout in the UNI-group after the first therapeutic intervention on the BTE Work Simulator. One study participant voluntarily ended participation in the study because of increased shoulder pain after the first therapy; thus, the initial examination data for this patient are omitted.

A defined treatment program was designed for both groups. Patients in the UNI-group performed uniplanar movements of the shoulder for flexion (lift arm forward), abduction (lift arm out to the side), and external rotation, while patients in the ADL-group practiced everyday movements, such as simulated climbing a ladder (alternate shoulder and elbow flexion and extension), bringing a glass to their mouth, and loading a shopping cart (Figure 1a,b). In a therapy session with a duration of approximately 25 min, the study participants performed the three exercises of the assigned group under the supervision of the therapist. The duration of the intervention was nine therapy units at the BTE over a period of three weeks. The total minutes of therapy completed in the rehabilitation center averaged 1961.11 ± 33.71 min in the UNI-group (n = 9) and 1976.50 ± 45.22 min in the ADL-group (n = 10).

### 2.2. Study Participants

Subjects between the ages of 40 and 70 years were enrolled in the study. The gender distribution at the initial examination was 60% females and 40% males in the ADL-group and 89% females and 11% males in the UNI-group. They all suffered from limited shoulder mobility and pain. The study participants were on average 55.1 years old (40–70 years) with a mean age in the UNI-group of 57.6 years (47–66 years, median 57.0) and in the ADL-group of 54.3 years (40–70 years, median 54.4). The mean BMI in the UNI-group was 26.6 (19.2–36.3, median 27.0) and in the ADL-group was 31.9 (26.6–42.2, median 32.7). Characteristics of participants at the beginning of rehabilitation are presented in Table 1.

### 2.3. Dash Questionnaire

The DASH is a 30-item questionnaire developed and validated in 1994 by the American Academy of Orthopedic Surgeons and others [34,35]. It is a patient-reported outcome measure (PROM) with a minimal clinically important difference set between 10.0 and 10.83 points and has good construct validity, test–retest reliability, and responsiveness to change [36,37]. This evidence has been provided for both proximal and distal disorders of the upper limb [38]. The DASH is composed of 30 questions, of which 21 questions relate to physical activity, such as writing or preparing a meal, six questions to symptoms, and three questions to social role. Subjects rate the symptoms or function of the upper extremity on a scale from one (no difficulties) to five (execution not possible). In this study, the DASH questionnaire was completed by the patients and calculated by the examiner. In addition, active and passive measurements of shoulder mobility using the neutral-zero method, as well as information on the pain situation at rest and in movement using numeric pain rating scale (NPRS), were noted. Functional examinations (data not shown) of the shoulder included Jobe, Hawkins, and Neer tests, an abduction resistance test, and recordings of the neck-grip (both hands behind the neck) and apron-grip (both hands behind the sacrum). Finally, the grip strength was measured using a dynamometer.

### 2.4. BTE-Settings

The Baltimore Therapeutic Equipment Work Simulator (BTE), developed in 1979 by Raymond Curtis and John Engalitcheff Jr., is a medically certified, computer-aided, commercial therapy device that enables the imitation of various movement patterns in everyday situations. The device consists of an exercise head providing a controlled resistance, to which various handles can be attached. Rehabilitation programs have applied the BTE for both testing and treatment purposes [39]. While standardized tests of physical fitness may not provide information on whether the subject is capable of performing a specific task that is essential for the particular occupation, the BTE is commonly used in occupational therapy for work evaluation and work-hardening programs [40,41].

In the UNI-group, flexion, abduction, and external rotation of the shoulder were trained with 3 times 15 repetitions per unit. The motor head of the Primus RS was mounted in a specific position, and the height of the pivot point was at the level of the glenohumeral joint or in line with the axis of rotation of the motor head. In the ADL-group, the motor head of the Primus RS was equipped with compatible tools. The tasks were trained with 3 times 15 repetitions per unit, and for climbing, as the movement was performed alternately, 3 times 30 repetitions.

### 2.5. Statistical Analysis

The data collected were analyzed using SPSS Statistics 21 (IBM) and Windows Excel 2016. A descriptive analysis of the basic data and calculations of the mean values were planned. Differences and anomalies should be identified and processed. The group data were compared before and after the start of the intervention and the significance level was set to *p* < 0.05. The evaluation of the metric data was carried out in different ways (difference calculation, mean comparison) due to the small number of cases. The data were then compared with one another. First, the Kolmogorov–Smirnov goodness-of-fit test was carried out to clarify the normal distribution with regard to the possible execution of the *t*-test. Then, the frequency calculation and a calculation using the *t*-test and Mann–Whitney U-test were performed for independent samples. Effect sizes were interpreted according to Cohen, while correlations between objective measures and NPRS and DASH were determined using Spearman’s rank correlation coefficients for t2 scores. In a subsequent step, the correlations between PROMs and performance measures within the respective groups were examined (Table 2). The relative hand grip strength (RHGS) was calculated by dividing maximum hand grip strength (HGS) by BMI.

## 3. Results

At the end of rehabilitation, the UNI-group showed a significant correlation between pain in movement and grip strength on the affected side. We also noted a significant negative correlation between the DASH and the aROM of abduction. On the other hand, we could not determine strong correlations between PROMs and objective performance scores in the ADL group.

### 3.1. DASH Results

At the beginning of rehabilitation (t1), the DASH ranged in the UNI-group between 7.50 and 59.16. As shown in Figure 2, the values at the end of rehabilitation (t2) were between 7.50 and 32.50 (mean value 20.27 ± 8.58, median 23.33). In the ADL-group, the DASH was at t1 between 19.16 and 62.50, and at t2 between 4.16 and 23.33 (mean value 12.91 ± 6.72, median 13.33). The mean value of the change (t2 − t1) of the DASH values amounted in the UNI-group to −10.92 ± 12.59, and in the ADL-group to −22.83 ± 11.31. All items in the UNI-group have improved, with the exception of item 16 (use a knife to cut food), which performed worse. Item 20 (manage transportation needs) and item 21 (sexual activity) showed no change. We observed the greatest positive change in item 6 (place an object on a shelf above your head). In the ADL-group, all items improved. The greatest improvement was recorded in item 12 (change a lightbulb overhead). A significant positive change in the DASH score could only be found in the ADL-group, which is significantly different to the UNI-group (*p* = 0.044).

### 3.2. Results Active Range of Motion (aROM)

We measured the active range of motion (aROM) for shoulder abduction of the affected and unaffected side during rehabilitation (Figure 3). The analysis of active mobility of the unaffected side revealed no significant group differences. On the affected side, aROM ranged in the UNI-group at t1 between 80–150° and at t2 between 90–150° (117.22 ± 20.48, median 110.0). The corresponding values in the ADL-group were at t1 80–160° and at t2 100–170° (139.50 ± 23.62, median 150.0). Between the beginning and the end of rehabilitation, we noticed abduction differences in the mean values in the UNI-group (2.77 ± 15.22) and in the ADL-group (25.50 ± 21.66). Both groups started with comparable values, but only the values of the ADL-group improved significantly (group difference: *p* = 0.018). This indicates that the exercise program in the ADL-group contributed to this improvement.

### 3.3. Pain (NPRS) Results

The results for pain are presented in Table 3, revealing no significant differences of relative changes in resting pain between the two groups (*p* = 0.270). For movement pain, we observed a significant improvement in the ADL-group compared to the UNI-group (*p* = 0.019). The UNI-group improved by 1.11 ± 2.37 points and the ADL-group improved by 3.70 ± 2.00 points. The ADL-group started with higher NPRS values. The initially different values for pain in motion can best be explained by the random group distribution and the small sample size; however, an improvement in the ADL-group could be verified both relatively and absolutely, nonetheless.

### 3.4. Results Grip Strength

The two group samples were inhomogeneous and showed a different distribution of gender and BMI. It was, therefore, to be expected that the values for the grip strength would also be distributed differently and a comparison of the results would be distorted [17,21]. We carried out correlation calculations between BMI and grip strength, which revealed intergroup differences. In the course of rehabilitation, however, both groups showed comparable intragroup behavior, which was also true for relative handgrip strength results, but the changes were not significant in either group. There were no significant differences in grip strength between the affected and the unaffected side. Grip strength of the affected side was enhanced in both groups. The values in the UNI-group only slightly improved from 29.67 ± 12.12 at t1 to 33.46 ± 12.72 at t2. In the ADL-group, the values improved from 20.92 ± 8.3 at t1 to 24.33 ± 6.40 at t2. In addition, RHGS showed at the beginning of rehabilitation a quotient of 1.10 in the UNI-group and 0.65 in the ADL-group and improved at the end of rehabilitation in the UNI-group to 1.27 and to 0.79 in the ADL-group. The intragroup differences were not significant (ADL-group: *p* = 0.40; UNI-group: *p* = 0.44).

## 4. Discussion

The shoulder complex connects the upper extremity with the trunk. The scapula, in particular, acts as a link between the trunk and arm, enabling the transmission and potentiation of strength, stability, and energy from the lower limbs and trunk to the upper extremity [42]. In shoulder rehabilitation, this relationship is increasingly understood and focuses on correcting posture [43], improving scapular control, and improving arm elevation [44]. This corresponds to our study design with complex ADL movements that involve several muscle groups not only in the shoulder girdle.

Traditionally, shoulder function has been assessed using clinical measures such as strength, pain, and range of motion. Advances in healthcare and the trend toward the use of PROMs have led to the proliferation of various such measures [45]. The present study shows the effects of training on the BTE Work Simulator on the DASH questionnaire and reveals significant differences in the group comparison, with a significant improvement in the ADL-group compared to the UNI-group in terms of pain in movement, DASH score, and shoulder mobility. We hypothesize that the complexity of movements and the adaptation of training units to daily life movement patterns contributed to this effect.

In a study with 85 subjects, Riebel et al. compared movement patterns (Selective Functional Movement Assessment, SFMA) with self-reported outcomes of different anatomical regions [46]. They detected fair to good positive correlations between improvements in self-reported outcomes and decreases in the number of painful patterns, consistent with the findings of the present study, in which improvement in patient-reported DASH scores correlated to some extent with pain reduction and better shoulder abduction.

Isokinetic training refers to exercises in which muscle contraction is performed with accommodating resistance throughout the active range of motion at a constant speed and enables subjects to contract skeletal muscles with near-maximal or maximal effort at controlled velocities [47,48]. Due to the uniplanar and strictly isokinetic movement patterns, certain training units on the BTE may not be adequate for real-life situations. This is supported by a study by Ting et al. where the measurement of bilateral lifting endurance on the BTE work simulator differed from actual lifting in a laboratory setting for healthy men [41]. Bélaise et al. identified both tangential and radial forces during shoulder movement, which were the result of a so-called misalignment angle between the longitudinal axis of the subject’s arm and the rotational axis of the dynamometer engine [49]. They assumed an underestimation of the flexion and abduction components of the shoulder torque measured by the isokinetic dynamometer, which probably contributed to an increased load on the shoulder joint.

In the present study, patients in the UNI-group showed worse mobility and pain levels than patients in the ADL-group did. This can partly be explained by different innervation patterns depending on angular velocities [47]. Another possible explanation is the involvement of different muscle groups in a particular movement. Wilk et al. suggested that training of the muscles of the shoulder girdle had to be combined with training of the trunk, the pelvic girdle, and the lower extremity to promote cognitive postural awareness, stability, and mobility in order to transfer energy from the upper to the lower extremity [50]. Comparable statements were made by Zinke and colleagues. In a study investigating the effects of isokinetic training on high-performance canoeists, their results revealed an interaction of the trunk and limb muscles, leading to better training results by improving stabilization and transferring forces and torque from the trunk to the upper limbs [51].

The primary aim was to attain a high level of group homogeneity based on the clinical history. The study participants should all have comparable limitations, but also a comparable rehabilitation potential. For this reason, patients were excluded according to the exclusion criteria mentioned above, resulting in a high number of excluded participants. Due to the small sample size, group homogeneity could be achieved according to clinical criteria, but not with regard to non-clinical characteristics. Nonetheless, the rehabilitation results were particularly clear for pain and limited mobility, thus further highlighting the positive effects of ADL training. The two groups were inhomogeneous and showed a different distribution of gender and BMI. It was, therefore, to be expected that the values for the grip strength would also be distributed differently, and a comparison of the results would be distorted [17,21]. We carried out correlation calculations between BMI and grip strength, which revealed intergroup differences. In the course of rehabilitation, however, both groups showed comparable intragroup behavior, which was also true for RGHS results.

It cannot be concluded from this that multiplanar movement exercises generally produce better results than uniplanar ones, since intermuscular and intramuscular balance ratios also need to be considered. Cools et al. compared the effect of twelve different shoulder exercises to determine which exercises are appropriate to optimize scapular muscle balance. They suggested the use of side-lying external rotation, side-lying forward flexion, prone horizontal abduction with external rotation, and prone extension exercises to promote activity in certain trapezius muscle parts [43]. Patients with glenohumeral disorders may exhibit imbalance within and between scapular muscles. To reestablish scapular mobility and stability, rehabilitation exercises focus on increasing strength and synchronous activation of the muscles that keep the shoulder in balance. In their study, Moeller et al. identified four exercises (bow-and-arrow, external-rotation-with-scapular-squeeze, lawnmower, and robbery) that meet these criteria optimally [52]. They assumed that multiplanar, multi-joint exercises represented movements that are more functional, which is in accordance with the finding of our study. In shoulder rehabilitation, exercises with complex movement patterns related to activities of daily living appear to lead to better rehabilitation outcomes.

This applies in a similar way to the behavior of pain levels over the course of rehabilitation. We observed improvements in both groups, but the improvement in the ADL group, in particular, was statistically significant. A distribution-based method determines the statistical extent of the change but does not provide any information about the clinical meaningfulness for patients. Michener and colleagues set the minimal clinically important difference (MCID) for shoulder pain on 2.17 points on the NPRS scale [53]. This was exceeded by the values for movement pain in the ADL-group and indicates the clinical importance of ADL training in shoulder rehabilitation.

In a study on lower extremity training, Stien et al. found that programs for multi-joint and single-joint training produced different outcomes [54]. Uniplanar single-joint movement patterns may lead to better stability, while multiplanar movements improve mobility. To regain mobility required for activities of daily living is an essential goal in shoulder rehabilitation. We, therefore, conclude that, tailored to specific indications, training programs with multiplanar movement patterns are the preferred instrument in shoulder rehabilitation.

### Limitations of the Study

The subjects suffered from various complaints of the shoulder girdle; the clinical pictures ranged from conservatively treated fractures to tendons tears and shoulder impingement or restricted mobility without specific diagnoses; however, the informative value of the study is reduced due to the small number of participants, group inhomogeneity, and the inhomogeneity of shoulder affections and initial pain scores.

## 5. Conclusions

The BTE Work Simulator enables the imitation of different isokinetic movement patterns in shoulder rehabilitation. Better scores on pain (NPRS mean values of pain in movement 3.6 vs. 4.1), active range of motion (mean values for shoulder abduction 139.50 ± 23.62 vs. 117.22 ± 20.48 degrees), and DASH score (12.91 ± 6.72 vs. 20.27 ± 8.58) indicate that specific exercise programs that imitate activities of daily living contribute to this improvement. We, therefore, conclude that, specifically for improving range of motion and reducing pain, multiplanar movement pattern training programs are the preferred tool in shoulder rehabilitation.

## Figures and Tables

**Figure 1 ijerph-19-10623-f001:**
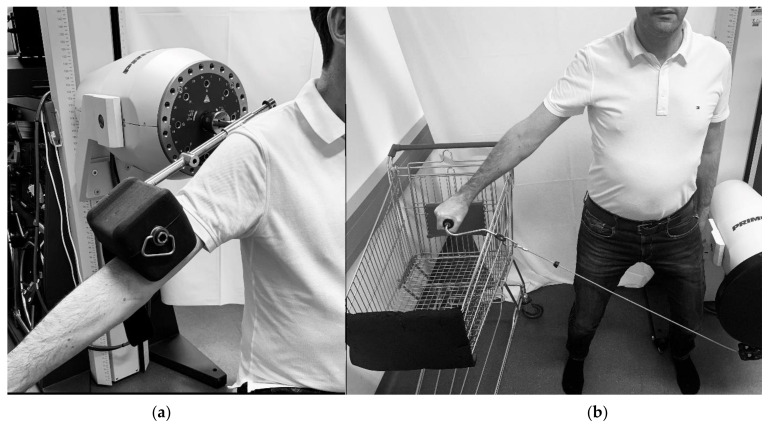
(**a**,**b**): Illustration of uniplanar and multiplanar movements: (**a**): uniplanar shoulder abduction; (**b**): multiplanar movement loading a shopping cart.

**Figure 2 ijerph-19-10623-f002:**
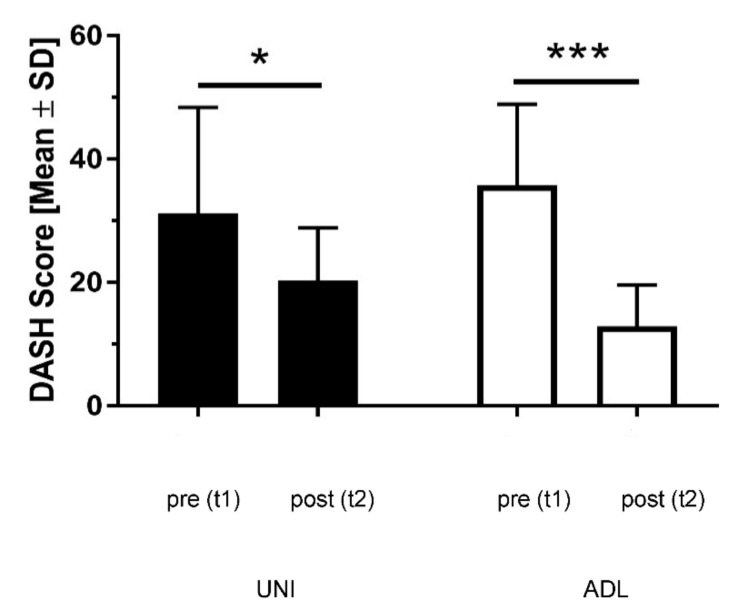
DASH score at t1 and t2 in the UNI-group and the ADL-group. Black columns: UNI-group; white columns: ADL-group; Asterisks indicating significance (*p* < 0.05 for * and *p* < 0.001 for ***); t1: beginning of rehabilitation; t2: end of rehabilitation; SD: standard deviation.

**Figure 3 ijerph-19-10623-f003:**
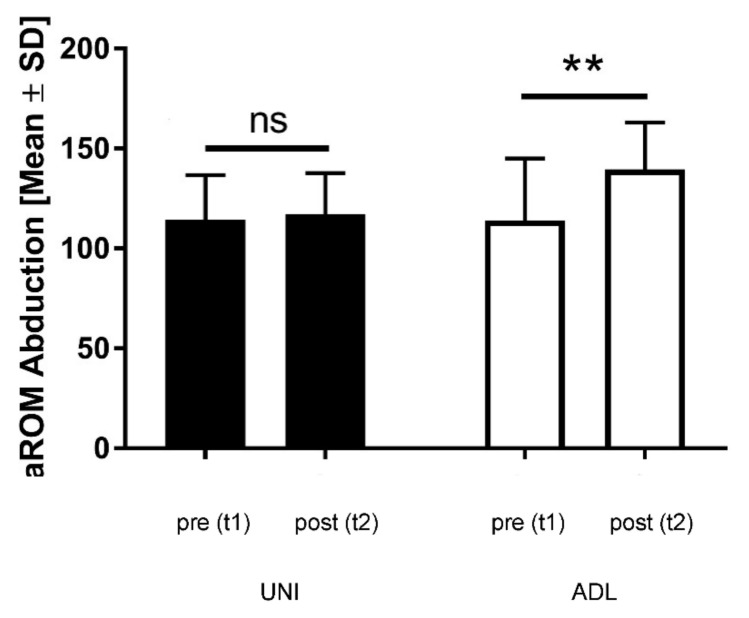
Shoulder abduction at t1 and t2 in the UNI-group and the ADL-group. Black columns: UNI-group; white columns: ADL-group; aROM: active range of motion; ns: not significant; Asterisks indicating significance (*p* < 0.01 for **); t1: beginning of rehabilitation; t2: end of rehabilitation; SD: standard deviation.

**Table 1 ijerph-19-10623-t001:** Characteristics of participants at the beginning of rehabilitation, mean values of BMI, age, DASH, pain (NPRS), hand grip, and aROM of the affected side.

Variable (SD, Median)	UNI-Group (n = 9)		ADL-Group (n = 10)		Total (n = 19)	
Male (%)	11		40		26	
Female (%)	89		60		74	
BMI	26.6	(±5.1, 27.0)	31.9	(±5.4, 32.7)	29.39	(±5.9, 28.5)
Age	57.6	(±5.9, 57.0)	54.3	(±9.5, 54.4)	55.1	(±8.1, 55.0)
DASH	31.2	(±2.2, 30.0)	35.8	(±1.9, 35.8)	33.6	(±2.4, 34.1)
NPRS at rest	1.8	(±2.5, 3.5)	2.6	(±1.6, 2.0)	2.2	(±1.2, 3.0)
NPRS in movement	5.2	(±1.9, 5.0)	7.3	(±2.0, 7.5)	6.3	(±2.1, 7.0)
HGS (N × 10)	29.7	(±12.1, 25.8)	20.9	(±8.3, 21.0)	25.5	(±11.36, 22.6)
RGHS	1.10	(±0.4, 1.0)	0.65	(±8.3, 0.4)	0.89	(±11.3, 0.9)
aROM abduction (degrees)	114.4	(±21.1, 120)	114.0	(±29.3, 105.0)	114.2	(±25.8, 110.0)

UNI-group: uniplanar movements; ADL-group: multiplanar movements; n = number; BMI: body mass index; DASH: disabilities of arm, shoulder, and hand score; NPRS: numeric pain rating scale; N: Newton; RGHS: relative hand grip strength; aROM: active range of motion; ±SD standard deviation and median in parenthesis.

**Table 2 ijerph-19-10623-t002:** Correlations of DASH, movement pain (NPRS), grip strength, and active ROM for abduction in the UNI-group (n = 9) and the ADL-group (n = 10) at the end of rehabilitation (t2).

		DASH		NPRS Move		Handgrip		aROM Abduction	
*Group*		*UNI*	*ADL*	*UNI*	*ADL*	*UNI*	*ADL*	*UNI*	*ADL*
DASH	r	1.000	1.000	0.551	0.555	0.286	−0.567	−0.775	0.211
	*p*	*-*	*-*	*0.124*	*0.096*	*0.456*	*0.087*	*0.014 **	*0.559*
NPRS move	r			1.000	1.000	0.790	−0.611	−0.470	−0.133
	*p*			*-*	*-*	*0.011 **	*0.061*	*0.201*	*0.714*
Hand grip	r					1.000	1.000	−0.462	−0.238
	*p*					-	-	*0.210*	*0.507*
aROM abduction	r							1.000	1.000
	*p*							-	-

DASH: disabilities of arm, shoulder, and hand score; NPRS: numeric pain rating scale; UNI-group: uniplanar movements; ADL-group: multiplanar movements; aROM: active range of motion; r: correlation coefficient; *p*: level of significance; Asterisks indicating significance.

**Table 3 ijerph-19-10623-t003:** Pain (NPRS) at rest and in movement, affected side.

Pain in Either Group.		Beginning of Rehabilitation (t1)	End of Rehabilitation (t2)	Relative Change (t1 − t2)	Significance of Change (*p*)
*NPRS at rest*					
UNI-group	Mean (SD)	1.78 (2.49)	1.11 (1.27)	−0.67 (2.50)	0.447
	Median (IQR)	0.0 (3.5)	1.0 (2.5)	0.0 (4.0)	
ADL-group	Mean (SD)	2.60 (1.58)	0.90 (1.29)	−1.70 (1.34)	0.003 **
	Median (IQR)	2.0 (2.3)	0.5 (1.3)	−1.5 (2.3)	
*NPRS in movement*					
UNI-group	Mean (SD)	5.22 (1.92)	4.11 (2.57)	−1.11 (2.37)	0.197
	Median (IQR)	5.0 (3.5)	4.0 (4.0)	0.0 (4.5)	
ADL-group	Mean (SD)	7.30 (2.00)	3.60 (2.55)	−3.70 (2.00)	0.000 ***
	Media (IQR)n	7.5 (2.8)	3.0 (4.3)	−4.5 (3.5)	

NPRS: numeric pain rating scale; UNI-group: uniplanar movements; ADL-group: multiplanar movements; SD: standard deviation; IQR: interquartile range; Asterisks indicating significance.

## Data Availability

The datasets analyzed in this manuscript are not publicly available, because of ethical and legal restrictions (data contain potentially identifying and sensitive patient information). If not already reported within this work, the authors may provide descriptive data on individual medical indicators for admission and discharge or the expected change due to inpatient health care for various groups and diagnoses. Requests for access to anonymized datasets should be directed to the corresponding author (vincent.grote@rehabilitation.lbg.ac.at).

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
