# Peer review of "Impact of Different Isokinetic Movement Patterns on Shoulder Rehabilitation Outcome"

_ijerph, 2022, doi:10.3390/ijerph191710623_

Round 1

Reviewer 1 Report

Thanks for this contribution.

Please note my suggestions to improve the clarity and understandability of your manuscript.

Please define/explain DASH score in abstract and as part of line 47.  It's not until line 57 when we finally find out what the DASH score entails.

Line 48-HG is an indicator of overall strength but does not measure overall strength.  Please rewrite.  Also, need more appropriate reference here.
Explain gender differences among groups (such as greater HG end of range for ADL.

Line 206 delete "of"

Line 212 delete "better"

Line 217 missing reference number and check at end references -there is a disconnect here.

Line 237 performing what?

Line 102-104--please need more details for both groups; perhaps an example of movement for flexion, abduction, and external rotation for both groups?

Please consider use of "elderly" throughout the manuscript. Individuals aged 40 to 70 are not "elderly"; Perhaps the use of "older adults" would be more respectful of advanced age without putting these older adults in a category that sounds like it's going "out to pasture."

Reviewer 2 Report

Impact of different isokinetic movement patterns on shoulder rehabilitation outcome

Abstract

It is necessary to add concrete data from the research that confirms the conclusions.

This is the description of the instrument and it should be in sector 2

It is necessary to add a chapter "Previous research", so that we can understand what makes this research different, that is, it contributes to the development of the recovery of patients with shoulder pain.

In the "Discussion" chapter, the authors simultaneously present previous research and discuss their results, which complicates monitoring the results.

A clearly defined research method is missing, which should be defined in the chapter "2. Materials and Methods". We assume that this is a study with two experimental groups, but this should be clearly presented.

Also, in the chapter "2. Materials and Methods" lacks representation of the sample by gender and age, which partially exists during the discussion of the results.

Until the end, it remained unclear whether there were 20 subjects in the sample, as written in the Abstract, or 9+10, as was later presented in the results.

Data from the initial measurement of both groups are missing. First, the data from the end of the study were presented.

202

3.4. Grip strength There were no significant differences in grip strength between the affected and the

unaffected side. Grip strength of the affected side enhanced in both groups. The values in the UNI-group only slightly improved from 30.18 ± 13.44 at t1 (range 13.33-53.0) to 33.92 ± 14.13 (range 16.67-50.33). In the ADL-group, the values improved from of 21.33 ± 8.4 at t1 (range 13.33-39.67) to 24.83 ± 6.59 at t2 (range 14.0-36.33).

In the handshake, the authors only compared raw data, which is not a correct approach, because we put all subjects on the same plane, regardless of their body mass, which differs at the start.

1. Body mass data should also be in Chapter 2.

2. We recommend to the authors that the hand grip is analyzed in relation to body mass, which is the standard set by the EUROFIT test.

3. The most serious methodological problem is that one group has a ratio of women to men of 60-40% and the other group 9 to 1, so it is impossible to compare them directly, because they are not equal in the beginning, nor through subsequent statistical procedures.

271 Conclusion

It is necessary to add concrete data from the research that confirms the conclusions.

Reviewer 3 Report

Dear author

Thank you for submitting to IJERPH. However, this study needs several modifications. Sorry, I have a hard time understanding this study. Sorry, I have a hard time understanding this study. This document is very weak in describing the research. Authors will be able to improve the document through clear section division, paragraph organization, figures, table representation, and logical introduction structure.

Reviewer 4 Report

Dear author

The study is significant and interesting for OTs and other health professionals treating shoulder disabilities. The idea of studying this issue is excellent, but most of my comments will be related to the presentation of the study. Herby are my comments:

1. The author should re-write the introduction. It is not sufficient. It is more about the research tools than the rationale of the study. You could re-write the introduction and include the following  information:

a. The reason for joint shoulder pain

b.How to treat shoulder pain. Which exercises are evidence-based?  Meaningful activities versus practices without meaning. Why should we think that ADL exercise will be more effective?

c. Please remove the description of the research tools from the introduction and re-write the methods.  

2. The author should present the participants and their age, the level of pain, and disability, separately, under the methods and not in the results.

3. a.The statistical analysis: The statistical analysis should include an explanation of the statistical methods. The results should be removed from this paragraph  to the Result section

b.The statistical analysis is primarily parametric.  In such a study sample size, it will be better to describe median and interquartile range instead of average and SD.

c.The correlation table should include both groups. Pls look at some published articles and how they present the results

d. You have to compare the groups before starting the treatment and check if they are equal in age, disability, gender 

Round 2

Reviewer 2 Report

The main problem of comparing two groups is their huge difference in gender and BMI results, so that all subsequent differences primarily come from differences in gender and BMI, which the authors do not take into account in their analyses.

Until the end, it remained unclear whether there were 20 subjects in the sample, as written in the Abstract, or 9+10, as was later presented in the results.

line 220

Intragroup differences correlating with the BMI were performed, instead of relativizing the grip strength values in relation to body mass, so that the relative hand grip strength can be seen.

Since there is no statistically significant difference there is no need to show the t-test values.

It is unclear why "Median" was inserted as a value, as it is not relevant to the calculation of the t-test and does not specifically indicate statistical differences between groups.

In Table 3, it is not clear why the obtained p value of 0.0223 is not statistically significant at the 0.05 level.

Reviewer 3 Report

Dear authors

The author has made many revisions. thank you. I hope to revise it a bit more for publication.

Detail is as attached file. 

Reviewer 4 Report

The author made significant progress while revising the article, but there are more revisions to make

The study design:

If it is a randomized control, write it according to the consort recommendation for RCTs

The study design is still not straightforward. Pls, write a paragraph about the participant's allocation. Write the inclusion and exclusion criteria and the participant's characteristics, and then describe the study procedure in a different paragraph

It would help if you changed the statistical analysis paragraph according to the changes that you will make.

The results:

Table 1:

A. What does the total mean? Is it the number(N) of participants? It is not clear. Please write the N and percent as follows: n(%)

B.NRPS, you cannot use mean if it is not normalllarge distributed. Us median and interquartile range.

Table 2:

A.pls. write at the bottom of the table the level of significance: p<0.05* and P<0.01 **

B. Pls write the R and P next to each other and not below. 

Table 3:

A . It is not enough to add the median. The author should add the interquartile range and omit the mean. The mean is not relevant here.

b.there is a reference about the minimal change in pain that has clinical meaning. If the mean is 1.76 and it became 1.11, does it have any clinical significance? The results showed a significant difference, but you should discuss its importance in the discussion.

Discussion:

 Pls explain the large amount of dropout you have

Pls relate to the level of pain and what it means 
